# Co-delivery of D-(KLAKLAK)_2_ Peptide and Chlorin e6 using a Liposomal Complex for Synergistic Cancer Therapy

**DOI:** 10.3390/pharmaceutics11060293

**Published:** 2019-06-21

**Authors:** Chaemin Lim, Jin Kook Kang, Woong Roeck Won, June Yong Park, Sang Myung Han, Thi ngoc Le, Jae Chang Kim, Jaewon Her, Yuseon Shin, Kyung Taek Oh

**Affiliations:** Department of Pharmaceutical Sciences, College of Pharmacy, Chung-Ang University, 221 Heukseok dong, Dongjak-gu, Seoul 06974, Korea; chaemin201@naver.com (C.L.); chief8631@gmail.com (J.K.K.); wonjas000@naver.com (W.R.W.); oong93@naver.com (J.Y.P.); hshsh0919@naver.com (S.M.H.); ngocle.dkh@gmail.com (T.n.L.); jchang1206@naver.com (J.C.K.); hjwon93@daum.net (J.H.); sus9417@hanmail.net (Y.S.)

**Keywords:** photo-chemo combination therapy, chlorin e6, pro-apoptotic peptide, liposomal complex, endosomal escape

## Abstract

Nanotechnology-based photo-chemo combination therapy has been extensively investigated to improve therapeutic outcomes in anticancer treatment. Specifically, with the help of a singlet oxygen generated by the photosensitizer, the endocytosed nanoparticles are allowed to escape from the endosomal compartment, which is currently an obstacle in nanotechnology-based anticancer therapy. In this study, a liposomal complex system (Lipo (Pep, Ce6)), composed of a chlorin e6-conjugated di-block copolymer (PEG-PLL(-*g*-Ce6)) and a D-(KLAKLAK)_2_ peptide loading liposome (Lipo (Pep)), was developed and evaluated for its anticancer activity. Due to the membrane lytic ability of the D-(KLAKLAK)_2_ peptide and the membrane disruptive effect of the singlet oxygen generated from chlorin e6, Lipo (Pep, Ce6) accelerated the disruption of the endosomal compartment, and exhibited strong synergistic anticancer activity in vitro. The prepared liposomal complex system could potentially maximize the efficacy of the nanotechnology-based photo-chemo combination therapy, and can be regarded as a novel, versatile strategy in advanced tumor therapy.

## 1. Introduction

Cancer is a disease with a high mortality rate. Current cancer therapies that rely on a single mode of treatment are often insufficient because of tumor heterogeneity or drug resistance, leading to frequent relapses [1,2,3]. To improve the therapeutic efficacy and minimize the side effects of the current treatment options, strategies that combine chemotherapy with radiotherapy, photothermal therapy, or photodynamic therapy have been extensively investigated [4,5,6,7]. Among these, the combination of chemotherapy with photodynamic therapy has emerged as a promising strategy for maximizing therapeutic efficacy. Photodynamic therapy (PDT) is based on the administration of a photosensitizer, followed by illumination of the tumor sites. In response to the illumination, the photosensitizer generates highly reactive oxygen species (ROS) that kill cancer cells without the risk of drug resistance [8,9,10,11]. Specifically, the ROS induced by PDT cause cell membrane lysis, resulting in an accelerated endosomal escape of the anticancer agent into the cytosol. As the escape of endocytosed nanoparticles from the endosomal compartment is a time-consuming step in nanotechnology-based anticancer treatment, the combination of PDT and chemotherapy is expected to maximize the therapeutic efficacy [12,13,14].

Biopharmaceuticals or bio-drugs have attracted much attention and have become “blockbusters” in the cancer chemo-drug market. Among these, the amphiphilic and cationic D-(KLAKLAK)_2_ peptide, first used as an antibacterial peptide, has been extensively investigated. The peptide forms an α-helical structure and induces mitochondria-dependent apoptosis. Therefore, the anticancer effect of this peptide has been investigated either alone, or in combination with other drugs [15,16].

A recent study reported that this peptide, when delivered in nanoparticles, could preferentially disrupt negatively charged cell membranes [17,18,19]. In addition, the combination of the formulated peptide with photodynamic therapy displayed powerful anti-bacterial activity due to the membrane lytic ability of the peptide and the generation of ROS by the photosensitizer [20]. Therefore, we hypothesized that when this combination is used in cancer treatment, the negatively charged organelle membrane in cancer cells could also be targeted, inducing cellular death pathways that are specific to cancer cells. In particular, the combination could help the endocytosed nanoparticles escape from endosomal compartment, which could improve the anticancer activity of pro-apoptotic peptide and photosensitizer. In the present study, we prepared a D-(KLAKLAK)_2_ peptide and chlorin e6 co-loaded liposomal complex system and evaluated its therapeutic potential using particle characterization and in vitro studies.

## 2. Materials and Methods

### 2.1. Materials

1,2-distearoyl-sn-glycero-3-phosphoethanolamine-*N*-[methoxy(polyethylene glycol)_2000_] (DSPE-PEG_2000_), Hydrogenated Soy l-α-phosphatidylcholine (HSPC), and cholesterol were purchased from Avanti Polar Lipids Inc. (Alabaster, AL, USA). Chlorin e6 (Ce6) was obtained from Frontier Scientific Inc. (Logan, UT, USA). Proapoptotic peptide D-(KLAKLAK)_2_ and FITC-labeled D-(KLAKLAK)_2_ were purchased from PepTron Inc. (Daejeon, South Korea). The dead cell apoptosis kit, the Caspase-3/7 Green detection reagent, and Lysotracker Red were purchased from Invitrogen. All other chemicals used were analytical grade. For cell culture, human cervical cancer KB cells were obtained from the Korean Cell Line Bank (KCLB, Seoul, South Korea). RPMI 1640 medium, fetal bovine serum (FBS), penicillin, and streptomycin were purchased from Welgene (Seoul, South Korea). The Cell Counting Kit-8 (CCK-8) was obtained from Dojindo Molecular Technologies (Tokyo, Japan).

### 2.2. Preparation of Blank Liposome and Drug-Loaded Liposome

Liposomes were prepared using the thin-film hydration method followed by sonication and extrusion, as previously reported [21]. Briefly, a mixture of HSPC:Chol:DSPE-PEG_2000_ (in a molar ratio of 55:40:5) was mixed in chloroform in a glass vial, and the organic solvent was removed using a rotary evaporator, followed by drying overnight in a vacuum desiccator. The lipid film was resuspended in distilled water, sonicated, and extruded 12 times through two stacked 200-nm nuclepore polycarbonate filters using a stainless steel extruder (Avanti Polar Lipids). For the peptide complexed liposome (Lipo (Pep)), the peptide solution was added to a lipid thin film at different weight ratios (4:1, 8:1, 16:1, and 32:1) and incubated for 4 h, followed by 12 rounds of extrusion.

### 2.3. Measurement of Size and Zeta Potential of Liposomes

The effective hydrodynamic diameter, size distribution, and zeta-potential of the prepared liposomes were measured by photon correlation spectroscopy using a Zetasizer Nano-ZS (Malvern instruments, Worcestershire, UK) equipped with the Multi Angle Sizing Option (BI-MAS). Each sample was stabilized for 30 min before measurement. The average particle size, polydispersity index (PDI), and zeta potential values were calculated from three independent measurements of each sample (*n* = 3) [21,22].

### 2.4. In Vitro Cellular Uptake Study

The uptake of peptide into KB cells was studied using flow cytometry. KB cells were seeded in 6-well plates at a density of 4 × 10^5^ cells/well. After 24 h incubation, each cell was exposed to FITC-labeled peptide (2.5 or 5 µg/mL of peptide) groups and incubated for 4 h. The cells were washed thrice with PBS, harvested, and analyzed in a FACSCalibur flow cytometer using Cell Quest Pro software (BD Biosciences, San Diego, CA, USA). To assess endosomal escape, KB Cells were seeded in a confocal dish at a density of 3 × 10^5^ cells/well, followed by an incubation period of 24 h. The cells were treated with each sample for 24 h and washed with PBS several times, followed by illumination at a light intensity of 5.2 mW/cm^2^ using a 670 nm laser source for 100 s. After 24 h, the cells were stained with Lysotracker Red DND-99 (100 nM) for 1 h, fixed with 4 % formaldehyde for 15 min, and stained with DAPI. Cells were observed by confocal microscopy (LSM 510 Meta, Carl Zeiss, Germany) [23,24].

### 2.5. Caspase 3/7 Activity Measurement

KB cells were seeded in 12-well plates at a density of 5 × 10^4^ cells/well, and exposed to free peptide or Lipo (Pep) (20 µg/mL of peptide) for 24 h. The cells were washed with PBS and stained with Caspase-3/7 Green Ready Probes ^TM^ Reagent (R37111, Invitrogen, Carlsbad, CA, USA) for 1 h to visualize the level of apoptosis. The stained cells were observed using Moticam Pro 205A camera coupled to a computer with the Motic Images Plus 2.0 (Richmond, BC, Canada) software.

### 2.6. Characterization of PEG-PLL(-g-Ce6)

The PEG-PLL(-*g*-Ce6) was synthesized by coupling PEG-PLL with activated Ce6 using DCC and NHS as previously reported by us [25]. Fluorescence images of PEG-PLL(-*g*-Ce6) or free Ce6 (equivalent Ce6 2 µg/mL) in PBS at different pH conditions were observed using a fluorescence-labeled organism bioimaging instrument (FOBI) fluorescence live imaging system (IFLIS; NeoScience, Suwon, South Korea). The UV-Vis spectra of PEG-PLL(-*g*-Ce6) or free Ce6 (equivalent Ce6 20 µg/mL) in DMF or distilled water were obtained using a UV-Vis spectrophotometer (Incheon, South Korea). For Ce6-complexed liposomes (Lipo (Ce6) or Lipo (Pep, Ce6), the blank liposome or peptide-complexed liposome was mixed with PEG-PLL(-*g*-Ce6) rapidly and vortexed immediately, followed by mild sonication. The amount of singlet oxygen generated by Lipo (Ce6) was measured using 9,10-dimethylanthracene as an extremely fast chemical trap for singlet oxygen [26,27]. The 9,10-dimethylanthracene (20 mmol) was added to each sample, followed by illumination of the solution at a light intensity of 5.2 mW/cm^2^ using a 670 nm laser source for 100 s. After 1 h, the change in fluorescence intensity (measured using a Scinco FS-2 fluorescence spectrometer at an excitation wavelength of 360 nm and emission wavelengths of 380–550 nm) was plotted for each sample after subtracting its fluorescence intensity in the absence of light illumination.

### 2.7. Cytotoxicity

KB cells harvested from growing cell monolayers were seeded in a 96-well plate at a density of 8 × 10^3^ cells per well, 24 h prior to the cytotoxicity test. Free peptide or Lipo (Pep) dispersed in RPMI 1640 medium were added to the cells followed by a 24-h incubation. Cells were washed with PBS and then further incubated for an additional 24 h. For the photo-toxicity test, each sample was treated and then incubated for 24 h. Cells were washed with PBS and illuminated at a light intensity of 5.2 mW/cm^2^ using a 670 nm fiber-coupled laser for 100 s and then incubated for 24 h. Cell viability was tested using a Cell Counting Kit-8 (CCK-8 assay). Combination index-Faction affected (CI-Fa) plots for evaluating the combined effects of pro-apoptotic peptide and photosensitizer were generated using Compusyn software (ComboSyn Inc., Paramus, NJ, USA). In the cell viability test, the survived cell in each treatment (0~100%) were converted to fraction affected number (Fa, 0~1) and Fa-CI plots were generated in each condition. CI values for the drug combinations are calculated according to the following equation:
$$\frac{\left(D\right)1}{\left(Dx\right)1}+\frac{\left(D\right)2}{\left(Dx\right)2}=CI$$
where (*D*)1 and (*D*)2 are the concentrations of photosensitizer and peptide in the combination resulting in fa growth inhibition, and (*D*x)1 and (*D*x)2 are the concentrations of the single drugs resulting in fa growth inhibition.

## 3. Results and Discussion

### 3.1. Preparation and Characterization of Liposomes

It is well known that the cellular uptake efficiency of multiple drugs in a single nanoparticle system is better than that of a mixture of several single drugs in different nanoparticle systems [25,28]; thus pointing to the importance of the physicochemical characteristics of drug carriers in combination therapy.

In this study, the conventional liposomes composed of HSPC, cholesterol, and DSPE-PEG (55:40:5 in molar ratio) were prepared to aid the formulation of the pro-apoptotic peptide and photosensitizer (Figure 1). Liposomes are known to have a small particle size with a slightly negative surface charge, which would help the formulation of the oppositely charged D-(KLAKLAK)_2_ peptide or PEG-PLL(-*g*-Ce6). The liposomes were prepared using the thin-film hydration method (Figure 2a) with continuous monitoring of the change of the particle size and PDI values over a seven-day period (Figure 2b). As expected, there were no significant changes in particle size and PDI, demonstrating the high stability of the prepared liposomes. Then, we prepared the D-(KLAKLAK)_2_ peptide loaded liposome system with different loading ratios (weight ratios) and measured the particle size, PDI, and zeta potentials (Table 1). When the positively charged D-(KLAKLAK)_2_ peptide was loaded into the liposome (Lipo (Pep)), there were no substantial differences in particle size and PDI values over the all peptide loading ratio. However, the zeta potential of Lipo (Pep) was considerably higher, demonstrating the electrostatic interaction between the prepared liposomes and the D-(KLAKLAK)_2_ peptide. However, when the loading content of the peptide in liposome was changed from 4:1 to 32:1, there was no significant difference in the zeta potential values, indicating that the peptide was mainly encapsulated in the inner space of the liposome.

### 3.2. In Vitro Characterization of Lipo (Pep)

The cellular uptake efficiency of drugs in nanoparticles could vary depending on the drug-loading ratio. In our study, we prepared a range of loading ratios of Lipo (_FITC-labeled_ Pep) (4:1, 8:1, 16:1, and 32:1 weight ratio) and used flow cytometry to examine the effects of different liposome-peptide complexation ratios on the cellular uptake of the peptide in KB cells.

As shown in Figure 3a, the free peptide could not easily penetrate the cell compartment due to its polarity [29]. However, when the peptide was complexed with a liposome, a notable enhancement of fluorescence intensity was observed, demonstrating the importance of nanoparticles in peptide delivery. In addition, when comparing the uptake efficiency of Lipo (Pep) at different loading ratios, the Lipo (Pep) (4:1) exhibited the highest fluorescence, possibly due to the high loading density of the peptide. These results strongly suggest that Lipo (Pep) (4:1) is the most promising formulation in terms of delivery efficiency and potential for further study.

The α-helical structure D-(KLAKLAK)_2_ peptide is known to induce mitochondria-dependent apoptosis [15,16]. Thus, the induction of apoptosis and necrosis by Lipo (Pep) (4:1) in KB cells was analyzed using a caspase-3/7 staining kit to obtain live images of the damaged KB cells (Figure 3b). When KB cells were exposed to each sample, only Lipo (Pep) displayed a strong fluorescence intensity, demonstrating the improved uptake efficiency and enhanced apoptosis signaling of the nanocarrier. The anticancer activity of free peptide or of Lipo (Pep) in KB cells was assessed using a cell viability test. Cells were exposed to a range of concentrations of D-(KLAKLAK)_2_ for 48 h, and the viability of KB cells was quantified using the CCK-8 assay. As shown in Figure 3c), the free peptide showed no remarkable toxicity in KB cells due to the low cellular uptake efficiency. However, the Lipo (Pep) achieved much better anticancer activities than the free peptide, demonstrating its potential for application in combination treatment.

### 3.3. In Vitro Characterization of Lipo (Ce6)

Under irradiation with a specific wavelength of light, the photosensitizer (Ce6) generates singlet oxygen, which is toxic to cancer cells. However, when the Ce6 molecules aggregate with each other because of hydrophobic interactions, the amount of singlet oxygen generated could dramatically decrease due to the self-quenching phenomenon, which is frequently observed in photodynamic therapy [30,31]. Thus, in the heterogeneous and complex in vivo environment, the loading of unaggregated photosensitizers into nanoparticles is preferable for retaining the efficacy of photodynamic therapy. Previously, we developed a PEG-PLL(-*g*-Ce6) (Di-Ce6) as an unaggregated photosensitizer for studying photo-chemo combination therapy [25]. In this study, the Di-Ce6 was prepared as a complex with oppositely-charged liposomes. The Ce6 was conjugated with the positively charged hydrophilic PEG-PLL di-block copolymer using DCC and NHS chemistry and confirmed by ^1^H NMR (Figure 4a). Due to the presence of three ionizable carboxyl groups in Ce6, the solubility of free Ce6 changes with the pH of the aqueous solution [32,33]. The free Ce6 aggregates or precipitates under acidic conditions, whereas the solubilized monomeric Ce6 is predominant with an increase in pH. However, the solubility of Di-Ce6 does not vary with pH, due to the hydrophilicity of the PEG-PLL di-block copolymer. As expected, the fluorescence intensity of free Ce6 was dramatically decreased in acidic conditions due to its low solubility (Figure 4b). However, there was no change in the fluorescence intensity of Di-Ce6 over the entire pH range studied.

The free Ce6 or Di-Ce6 was dispersed in distilled water (the pH of distilled water was about 5.1) at the same Ce6 concentration (20 μg/mL) and its UV/Vis spectrum was investigated. As shown in Figure 4c, due to the low solubility of free Ce6, almost all the free Ce6 were precipitated, resulting in self-quenching and low absorbance spectrum, whereas the Di-Ce6 exhibited high absorbance due to its increased water solubility. Together, these results demonstrate that the Ce6 was successfully conjugated with the lysine residue in the PEG-PLL di-block copolymer. The prepared Di-Ce6 has a positive charge due to the lysine residue, allowing it to be complexed with the negatively charged liposome via electrostatic interaction. Figure 4d shows the particle size, PDI, and zeta potential of the liposome/Di-Ce6 complexes (Lipo (Ce6)) at different mixing ratio. When the Di-Ce6 was complexed with the liposome (Lipo (Ce6)), there were no remarkable differences in particle size and PDI values over the entire range of complexation ratios. However, as the amount of added Di-Ce6 increased, the zeta potential of Lipo (Ce6) dramatically increased to +20 mV, demonstrating the electrostatic interaction between the prepared liposomes and Di-Ce6. Under light illumination, the Lipo (Ce6) generated singlet oxygen in a Ce6 dependent-manner (Figure 4e)**.** It also presented substantial anticancer activity against KB cells (Figure 4f), demonstrating the feasibility of a liposomal complex as a PDT agent for anticancer treatment.

### 3.4. In Vitro Evaluation of Lipo (Pep, Ce6)

As the photosensitizer and the apoptotic peptide can act synergistically to effect bacterial membrane lysis [20,34], the combination of Lipo (Ce6) and Lipo (Pep) could potentially be used to achieve a similar synergistic effect in anticancer treatment. The in vitro characterization indicated that the Lipo (Ce6) and Lipo (Pep) may have considerable potential for application in anticancer therapy. Thus, we prepared a Ce6 and peptide co-loaded liposomal complex system (Lipo (Pep, Ce6)) and evaluated its anticancer activity in KB cells. The Lipo (Pep) (4:1), which exhibited the highest peptide delivery efficiency in vitro, was complexed with Di-Ce6 at different drug combination ratios. The particle size, PDI, and zeta potential values of the prepared Lipo (Pep, Ce6) were investigated (Figure 5(a)). Corroborating our earlier results (Figure 4d), there were no notable differences in particle size and PDI values over the entire range of Pep/Ce6 complexation ratios.

However, as the amount of added Di-Ce6 increased, the zeta potential of Lipo (Ce6) increased to +7 mV, demonstrating the electrostatic interaction between the Lipo (Pep) and Di-Ce6, resulting in Lipo (Pep, Ce6). Then, we investigated the endo-lysosomal membrane lytic ability of Lipo (Pep, Ce6) in KB cancer cells. There were several reports that the combination of photosensitizer and antimicrobial peptide could disrupt the cellular membrane efficiently. When the photosensitizer and antimicrobial peptide is applied to cancer therapy, it is also expected to disrupt negatively charged cellular membranes efficiently. Especially given that the failure to escape from the endo-lysosomal compartment of endocytosed nanoparticles is the biggest challenges in cancer therapy [35,36,37], the combination of antimicrobial peptide and photosensitizer is anticipated to be of potential use for maximizing therapeutic efficacy. To determine whether or not endosomal membrane lysis was occurring because of Lipo (Pep, Ce6), KB cells were exposed to each sample and stained by Lysotracker dye under light illumination (Figure 6a). Untreated cells and those that were treated with blank liposomes displayed endosomal compartments that were stained strongly with the Lysotracker dye. Cells displayed a slightly decreased intensity of the Lysotracker when exposed to Lipo (Pep), and a dramatic decrease when exposed to Lipo (Pep, Ce6). These results point to the accelerated lysis of the endosomal membrane in response to antimicrobial peptide/photosensitizer combination therapy, which could contribute to the enhanced anticancer activity of D-(KLAKLAK)_2_ peptide and chlorin e6.

### 3.5. Anticancer Activity of Lipo (Pep, Ce6)

It is well known that the D-(KLAKLAK)_2_ can induce mitochondria-dependent apoptosis. Thus, when the formulated D-(KLAKLAK)_2_ escapes the endo-lysosomal compartment more efficiently due to the effect of Lipo (Pep, Ce6) under light illumination, the peptide is more accessible to the mitochondria and has better anticancer activity. The in vitro anticancer activities of Lipo (Pep, Ce6) prepared with different drug loading ratios were compared using a cell viability test (Figure 6b).

At various concentrations of peptides ranging from 0.625 to 10 µg/mL, Lipo (Pep) was insufficiently cytotoxic to KB cells, whereas all concentrations of Lipo (Pep, Ce6) exhibited enhanced anticancer activity due to the co-formulated Ce6 in the liposome. To quantify the efficacy of combination therapy, combination index-affected factors (CI-Fa) plots for various combinations of peptide and Ce6 were generated (Figure 6b). When cells were treated with Lipo (Pep, Ce6), the CI values were much lower than 1 for the 10:1, 10:0.5, and 10:0.25 drug combination ratios, which clearly demonstrates the synergistic effect of the peptide and the Ce6 in the liposome system. The CI values of Lipo (Pep, Ce6; 10:1) were slightly lower than those of the 10:0.5 and 10:0.25, which may be due to the surface charge of the liposome and the tight complexation of the Ce6 to the liposome system. As control groups, the cell viability of Lipo(Pep, Ce6) at different complex ratio without light illumination was also investigated in Appendix A. As the Ce6 can be activated only under the light illumination, there were no significant differences in the cell viability between Lipo(Pep) and Lipo(Pep, Ce6) groups.

We hypothesized that the formulated peptide will escape from the endosomal compartment efficiently due to the membrane-lytic ability of the peptide and the photo-chemical internalization effect induced by Ce6 under light illumination. We demonstrated that the endosomal compartment was disrupted and that anticancer activity was dramatically enhanced when the peptide and Ce6 were combined in the liposome. From these results, we conclude that the observed strong synergistic effect of the peptide/Ce6 combination treatment was related to the escape of the dual drug-loaded liposome from the endosomal compartment under light illumination, and that it may have considerable potential for in vivo use. Thus, the combination of D-(KLAKLAK)_2_ and Ce6 could potentially be used to maximize the efficacy of anticancer therapy. When we started this project, we believed that the Pep/Ce6 combination would have considerable potential for use in anticancer treatment against various cancer cells. Thus, we chose the KB cells as one of the most common cancer cells in scientific research area.

## 4. Conclusions

Recently, photo-chemo combination therapy using D-(KLAKLAK)_2_ peptide and photosensitizer was reported as displaying strong anti-bacterial activity due to the membrane lytic ability of the peptide and the photosensitizer’s ability to generate ROS. We hypothesized that when the above combination is applied to cancer cells, the negatively charged organelle membranes in cancer cells could also be severely damaged, inducing apoptosis. Thus, the D-(KLAKLAK)_2_ and chlorin e6 co-loaded liposomal complex system was prepared as a photo-chemo combination therapeutic agent and evaluated for potential use in anticancer treatment. The pro-apoptotic peptide was loaded into the liposome and PEG-PLL(-*g*-Ce6) were complexed with Lipo (Pep), resulting in Lipo (Pep, Ce6). The formulated peptide successfully entered the cells and exhibited improved anticancer activity. Under light illumination, the prepared formulation generated singlet oxygen species and accelerated endosomal membrane lysis, which is the most important challenge in nanotechnology-based anticancer treatment. The Lipo (Pep, Ce6) showed enhanced anti-proliferative activity in vitro, with strong synergistic effects. Thus, this novel combination has considerable potential for use against cancer cells and can be regarded as a versatile strategy in advanced tumor therapy.

## Figures and Tables

**Figure 1 pharmaceutics-11-00293-f001:**
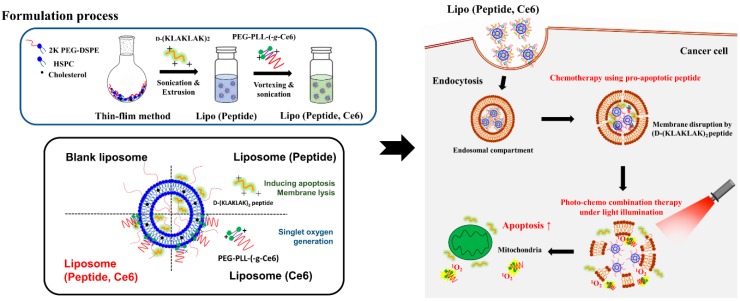
Schematic conceptual representation of the pro-apoptotic peptide and photosensitizer co-loaded liposomal complex system.

**Figure 2 pharmaceutics-11-00293-f002:**
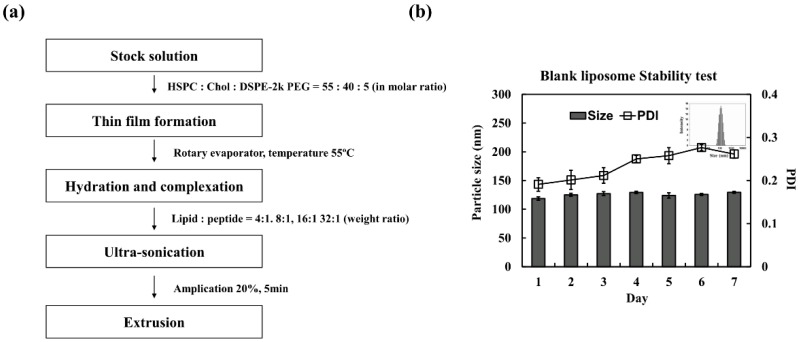
(**a**) Preparation method of liposome and (**b**) stability of the prepared liposome in saline.

**Figure 3 pharmaceutics-11-00293-f003:**
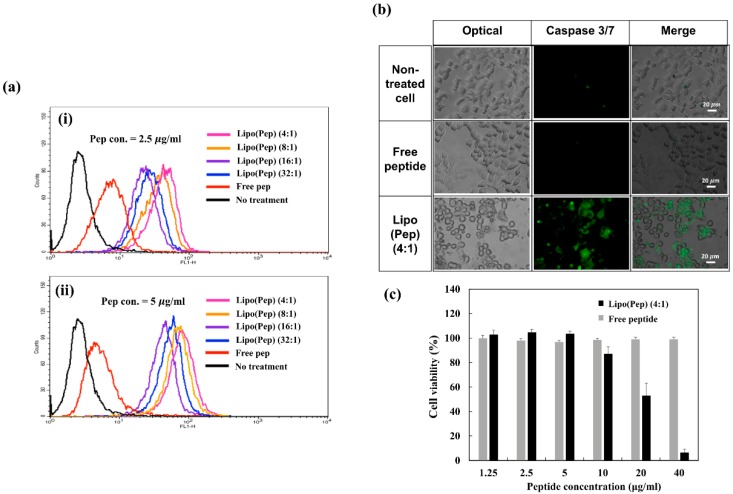
In vitro characterization of free peptide and Lipo (Pep); (**a**) quantification of cellular uptake of FITC-labeled peptide in KB cells at different peptide concentration; (**b**) effect of peptide on caspase 3/7 activity in KB cells; (**c**) cell viability of KB cells treated with Lipo (Pep) or free peptide.

**Figure 4 pharmaceutics-11-00293-f004:**
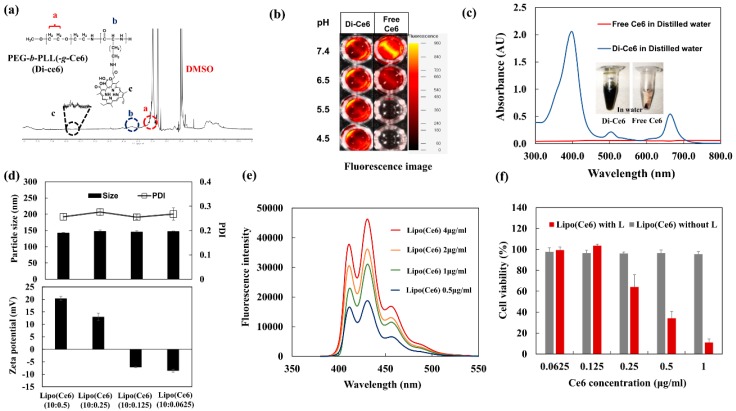
In vitro characterization of Lipo (Ce6); (**a**) ^1^H NMR spectra of PEG-PLL(-*g*-Ce6) (Di-Ce6); (**b**) fluorescence image of Di-Ce6 and free Ce6 in PBS at different pH conditions; (**c**) UV-Vis spectrum scan of free Ce6 and Di-Ce6 in distilled water; (**d**) particle size, PDI and zeta potential of Lipo (Pep) at different loading ratio (*w*/*w*); (**e**) the 9,10-dimethylanthracene fluorescence change of Lipo (Ce6) at different Ce6 concentration (5.2 mV, 100s laser exposure time); (**f**) the cell viability of KB cells treated with Lipo (Ce6) or free Ce6 (5.2 mV, 100s laser exposure time).

**Figure 5 pharmaceutics-11-00293-f005:**
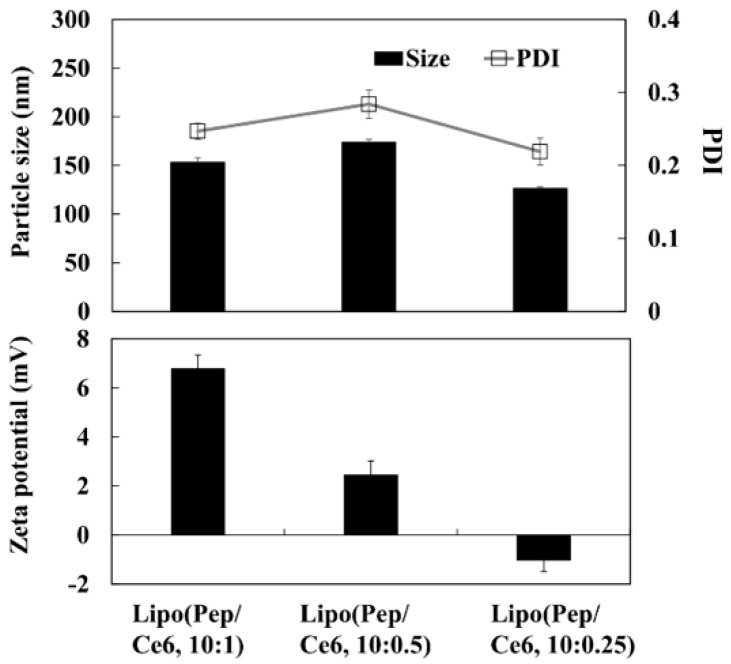
Particle size, PDI and zeta potential of Lipo (Pep, Ce6) at different drug loading ratio (*w*/*w*).

**Figure 6 pharmaceutics-11-00293-f006:**
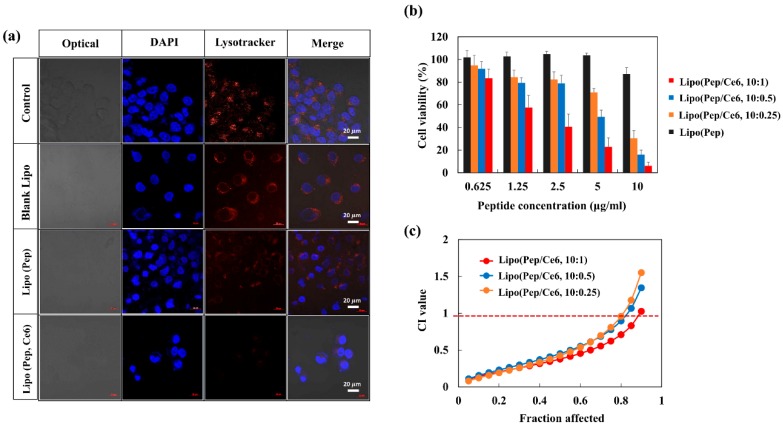
(**a**) Optical and fluorescence images of KB cells treated with blank liposome, Lipo (Pep) or Lipo (Pep, Ce6). (**b**) The cell viability of KB cells treated with Lipo (Pep) or Lipo (Pep, Ce6) at different drug loading ratio (5.2 mV, 100 s laser exposure time). (**c**) Plot of CI values for combination therapy at different peptide/Ce6 mixing ratio.

**Table 1 pharmaceutics-11-00293-t001:** Characterization of peptide complexed liposomes.

Polymer	Size (nm) ^a^	PDI ^a^	Zeta potential (mV) ^a^
**Blank liposome**	118.7 ± 2.8	0.19 ± 0.02	−10.6 ± 0.8
**Liposome/peptide (4:1)**	130.1 ± 3.2	0.22 ± 0.02	−3.3 ± 0.9
**Liposome/peptide (8:1)**	111.6 ± 1.0	0.15 ± 0.03	−4.1 ± 0.3
**Liposome/peptide (16:1)**	124.0 ± 0.6	0.19 ± 0.04	−5.8 ± 0.4
**Liposome/peptide (32:1)**	126.2 ± 1.8	0.18 ± 0.02	−4.7 ± 0.3

^a^ Size and zeta potential were measured using a Zetasizer Nano-ZS (Malvern instruments, UK) equipped with the Multi Angle Sizing Option (BI-MAS).

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
