# Peer review of "Co-delivery of D-(KLAKLAK)2 Peptide and Chlorin e6 using a Liposomal Complex for Synergistic Cancer Therapy"

_pharmaceutics, 2019, doi:10.3390/pharmaceutics11060293_

Round 1

Reviewer 1 Report

The manuscript proposed by Kyung Taek Ohand and coll. (Ref. No.: pharmaceutics-509402) for publication in Pharmaceutics is entitled “Co-delivery of D-(KLAKLAK)2 peptide and chlorin e6 using a liposomal complex for synergistic cancer therapy”. In this study the authors develop the use of a liposomal complex system (Lipo (Pep, Ce6)), composed of a chlorin e6-conjugated di-block copolymer [PEG-PLL(-g-Ce6), chlorin being the photosensitizer, PS] and a D-(KLAKLAK)2  peptide loading liposome (Lipo (Pep)), and they evaluated for its anticancer activity. Due to the membrane lytic ability of the D-(KLAKLAK)2  peptide and the membrane disruptive effect of the singlet oxygen generated from PS, (Pep, Ce6) accelerated the disruption of the endosomal compartment, and seems to exhibit a synergistic anticancer activity in vitro. The authors claimed that the prepared liposomal complex system could potentially maximize the efficacy of the nanotechnology-based photo-chemo combination therapy and can be regarded as a novel strategy in advanced PDT.

Few comments, questions and remarks

- The characterization of the cytotoxicity and the PHOTO-cytotoxicity (and related comparison) is not clear all along the manuscript; to the best of my lecture, knowledge, the cytotoxicity of the (Pep, Ce6) is not described; an the authors comment on that? Of course it seems to be more photo-cytotoxic in presence of an already well-documented photosensitizer.

- Idem, the potential application in cancer therapy and/or anti-bacterial activity is not clear; all along the manuscript, anti-cancer activity is claimed (“3.5 Anticancer activity of Lipo (Pep, Ce6)”) and in the conclusion a “strong anti-bacterial activity due to the membrane lytic ability” is more pointed out ; it is not clear.

- I am very surprised that the “zeta potential were measured using DLS equipment”

To conclude, this manuscript may be of interest, but the presentation and discussion of the results are not clear enough. My conclusion is that the authors have to re-write and comment more precisely, BEFORE this manuscript could be re-evaluated (re-submitted) for potential publication in pharmaceutics.

Author Response

Dear Reviewer 1, 

We appreciate the reviewer's valuable comments. We attached our revision report and revised manuscript. All changes are highlighted in the revised manuscript. 

Thank you very much

Reviewer 2 Report

Lim and co-workers present a liposome-based Co-delivery of D-(KLAKLAK)2 peptide and chlorin e6, which shows a synergistic cancer therapy effect in vitro. The manuscript is generally well written and the data presentation is clear. This photo-chemo combinations approach may suggest a way to enhance the efficacy cancer treatment. Here are some comments to be addressed.

1. According to the procedure of liposome preparation (extrusion at last), it is not surprised that the size of liposomes did not vary a lot, as long as the nanoparticles are stable. However, the change in zeta potential seem interesting and worth discussing. For example, line 156, “there was no significant difference in the zeta potential values”. In fact, there is a change in zeta potential after complexed with the peptides (Table 1, from -10 to -4 mV). Please explain.

2. Is the peptide D-(KLAKLAK)2 positively or negatively charged? Please describe and discuss with the change in zeta potential in the liposome-peptide complex.

3. In Figure 4c, what is the pH of distilled water and are the Di-Ce6 and Free Ce6 in a same concentration? It seems the Free Ce6 aggregated before the UV-vis measurement. Will a sonication make the comparison fairer? Interestingly, in pH 6.5 and 7.4, the Free Ce6 gave a strong fluorescence comparable to the Di-Ce6 (any aggregation in the fluorescent imaging?). Please comment.

4. The study presents the cell viability tests on liposome-peptide, liposome-peptide-Ce6, liposome-Ce6, and demonstrates the individual and synergistic cytotoxicity of the peptide and Ce6. Herein, Ce6 was activated by light illumination. As a control, it would be interesting to know the cell viability results of liposome-peptide-Ce6, and liposome-Ce6 without light illumination, and compare the present data.

5. Please describe the rationale of using KB cell as a model. What type of cancer does it represent for?

6. Line 256, “Figure 5(b)”, should it be Figure 6(a)? please double check the figure number.

7. Line 268, “Figure 6(a)”, should it be Figure 6(b)? please double check the figure number.

8. Line 275, please describe in Method section on how to calculate combination index-affected factors (CI-Fa).

9. To make the results easily repeatable for readers, some experimental details need to be clarified. For example, Line 80, “molecular weight ratios”, does it mean weight ratios. Line 91 “The next day”, please state the exactly time (hours) for cell grow?

Author Response

Dear Reviewer 2, 

We appreciate the reviewer's valuable comments. We attached our revision report and revised manuscript. All changes are highlighted in the revised manuscript. 

Thank you very much

Round 2

Reviewer 1 Report

First concerning my remark on the lake of study (corresponding results) on the cytotoxicity of the (Pep, Ce6), I don’t agree with the authors; this was NECESSARY; it is not because the PHOTO-cytotoxicity of a PS is actually activated by the light that its CYTOTOXICITY is not possible; This seems to have been added; we may trust these necessary data.

YES, the sentence “ Size AND ZETA POTENTIAL were measured using DLS equipment “ was a mistake that has been corrected.

Anti-cancer activity has been clarified (which was really necessary in the previous version)

I think now this manuscript deserve for publication in Pharmaceutics.